# End-User Assessment of an Innovative Clothing-Based Sensor Developed for Pressure Injury Prevention: A Mixed-Method Study

**DOI:** 10.3390/ijerph20054039

**Published:** 2023-02-24

**Authors:** Anderson S. Rêgo, Luísa Filipe, Rosana A. Dias, Filipe S. Alves, José Queiroz, Alar Ainla, Luísa M. Arruda, Raul Fangueiro, Maria Bouçanova, Rafael A. Bernardes, Liliana B. de Sousa, Paulo Santos-Costa, João A. Apóstolo, Pedro Parreira, Anabela Salgueiro-Oliveira

**Affiliations:** 1Health Sciences Research Unit: Nursing (UICISA: E), Nursing School of Coimbra (ESEnfC), 3000-232 Coimbra, Portugal; ltrfilipe@esenfc.pt (L.F.); rafaelalvesbernardes@esenfc.pt (R.A.B.); baptliliana@esenfc.pt (L.B.d.S.); paulocosta@esenfc.pt (P.S.-C.); apostolo@esenfc.pt (J.A.A.); parreira@esenfc.pt (P.P.); anabela@esenfc.pt (A.S.-O.); 2International Iberian Laboratory of Nanotechnology (INL), 4715-330 Braga, Portugal; rosana.dias@inl.int (R.A.D.); filipe.alves@inl.int (F.S.A.); jose.queiroz@inl.int (J.Q.); alar.ainla@inl.int (A.A.); 3Fibrenamics, Institute of Innovation on Fibre-Based Materials and Composites, University of Minho, 4800-058 Guimaraes, Portugal; luisa.arruda@fibrenamics.com (L.M.A.); rfangueiro@det.uminho.pt (R.F.); 4Centre for Textile Science and Technology (2C2T), University of Minho, 4800-058 Guimaraes, Portugal; 5Impetus Portugal-Têxteis Sa (IMPETUS), 4740-696 Barcelos, Portugal; mboucanova@impetus.pt

**Keywords:** biomedical technology assessment, wearable electronic devices, protective clothing, pressure injury, mobility limitation, bedridden persons

## Abstract

This study aimed to evaluate a clothing prototype that incorporates sensors for the evaluation of pressure, temperature, and humidity for the prevention of pressure injuries, namely regarding physical and comfort requirements. A mixed-method approach was used with concurrent quantitative and qualitative data triangulation. A structured questionnaire was applied before a focus group of experts to evaluate the sensor prototypes. Data were analyzed using descriptive and inferential statistics and the discourse of the collective subject, followed by method integration and meta-inferences. Nine nurses, experts in this topic, aged 32.66 ± 6.28 years and with a time of profession of 10.88 ± 6.19 years, participated in the study. Prototype A presented low evaluation in stiffness (1.56 ± 1.01) and roughness (2.11 ± 1.17). Prototype B showed smaller values in dimension (2.77 ± 0.83) and stiffness (3.00 ± 1.22). Embroidery was assessed as inadequate in terms of stiffness (1.88 ± 1.05) and roughness (2.44 ± 1.01). The results from the questionnaires and focus groups’ show low adequacy as to stiffness, roughness, and comfort. The participants highlighted the need for improvements regarding stiffness and comfort, suggesting new proposals for the development of sensors for clothing. The main conclusions are that Prototype A presented the lowest average scores relative to rigidity (1.56 ± 1.01), considered inadequate. This dimension of Prototype B was evaluated as slightly adequate (2.77 ± 0.83). The rigidity (1.88 ± 1.05) of Prototype A + B + embroidery was evaluated as inadequate. The prototype revealed clothing sensors with low adequacy regarding the physical requirements, such as stiffness or roughness. Improvements are needed regarding the stiffness and roughness for the safety and comfort characteristics of the device evaluated.

## 1. Introduction

Quality of life is significantly impacted by pressure injuries (PIs), which are an important risk factor for morbidity and mortality rates, namely among bedridden people and/or people with reduced mobility (RM) [1,2,3]. Alongside the impact on patient outcomes and care experience, PIs account for a significant socioeconomic burden with increased admission periods and an increase in the complexity of care and related costs [3,4].

PIs usually affect tissue composition in extension and depth. They progress rapidly and are caused by pressure, friction, and shearing forces from the continuous use of materials such as mattresses, sheets, clothing, and medical devices, with greater incidence risk in anatomical regions with bony prominences [1,2,3].

Epidemiological data highlight a pooled prevalence of PIs of approximately 21% worldwide [4]. In Europe, millions of people develop this clinical condition, with a prevalence ranging from 7% to 23% [3]. The cost of treating PIs is high. In the United States of America, treatments can exceed USD 20 billion, of which 58% is only for late-stage PIs [5]. In Australia, hospital-acquired PIs account for over US$ 9 million spent on treatments [6]. In this sense, the development and implementation of preventive and therapeutic interventions is a complex challenge that needs to be addressed [4,5,6].

Sensor-based medical devices are one of the most recent technological advancements in this area and have shown to be effective in assisting healthcare professionals during decision making [7,8]. In this perspective, the research related to new technology aimed at the development of new materials, such as highly sensitive sensors for clinical sign monitoring and adjusted to the skin microclimate, has been a common growing interest in the scientific community and medical industry due to their high performance. Hence, the development of functional textiles for monitoring PI risk factors usually focuses on durability, low cost, and accuracy of the parameters to be assessed, despite the highly complex fabrication process [4,7,9]. Current trends have been focused on multilayered textiles, with the potential to redistribute pressure in body regions [10]. Additionally, either integrated or externally attached sensors have been reported as valuable alternatives in terms of permeability, feasibility, and adaptation to body surfaces, suitable to distinct clinical contexts [11].

In this context, the project “4NoPressure: Development of smart clothing for pressure ulcer prevention” aims to research and develop clothing under the Smart Health Textiles typology for bedridden people and/or people with RM to reduce the occurrence of PIs. This project also proposes the real-time monitorization of specific clinical indicators in people at a high risk of developing PIs in hospital environments, receiving long-term care, or receiving in-home care. Moreover, it proposes an effective device for clinical decision making, thus contributing to improving the quality of life of these people [12,13,14,15].

Integrative approaches are envisaged, including sensor-based clothing to prevent PIs and the necessary personalized management to ensure patient safety. Due to the instability of the biological system, any medical device development may be a potential risk factor for PIs when in contact with the skin, so it is important to promote the analytical performance of new wearable devices [16,17,18].

Within this research topic, international experts have prioritized specific methodological strategies related to the development of medical devices, such as smart clothing technologies [19]. Major limitations on this type of research arise from the fact that wearable sensors have levels of complexity that go beyond the components required for their performance in detecting certain stimuli of clinical interest. Mechanical properties must be observed, considering flexibility, elasticity, and strength [20,21].

The project reported in this paper focuses on Technology Readiness Levels (TRLs) 1 and 2. These levels are specific for the relative exploration of the basic principles and concept testing to obtain the adjustments between the technology to be developed and its final production [21,22,23,24,25]. To this end, feedback from primary or secondary end-users can inform the iteration process, ensuring a design tailored to their preferences and needs [26,27,28,29]. This can also be called “user-centered design”, which considers human factors in research, such as comfort, ergonomics, aesthetics, and general usability issues, thus aiming for a device that is not only functional but also attractive and pleasant to use [26,27,28,29,30,31,32].

The mentioned methods directed to smart clothing sensors and their development have a scarce presence within the scientific literature. Thus, this study aims to ensure that the device under development meets the needs and preferences of its final users, making it more likely to be used consistently and effectively and thus contributing to user safety [33,34,35,36,37].

Thus, this research aimed to evaluate the physical and comfort requirements of a prototype of pressure, temperature, and humidity sensors to be integrated into clothing to prevent the occurrence of PIs.

## 2. Materials and Methods

### 2.1. Study Design

A mixed-method study was conducted using concurrent data triangulation [38] and following the guidelines of the Mixed Method Appraisal Tool (MMAT) [39] (Figure 1).

The mixed method was used to complement the topic under study and determine the main convergences, divergences, and integration between the quantitative and qualitative assessments conducted [38,39].

### 2.2. Study Setting

The study was developed at the Health Sciences Research Unit: Nursing (UICISA: E), and data were collected at the facilities of the Nursing School of Coimbra (ESEnfC), Portugal. The stakeholder meeting and sequential methodological implementation (QUAN + QUAL) were carried out to pursue the planned strategy [38].

### 2.3. Participants and Recruitment

For the evaluation process, nursing professionals with knowledge, skills, and experience in caring for people with RM and/or bedridden people were recruited. Eligible participants were those nursing professionals who presented at least two of the following criteria: research on the development of preventive and/or health promotion actions aimed at people with RM and/or bedridden people [40].

Nursing professionals were recruited to participate through an active search for contacts and invitation letters to establish a convenience sample. Nine nursing professionals active in teaching, research, and/or assistance, in the areas of rehabilitation, intensive care, long-term care, general nursing, oncology, and medical or surgical settings, participated in the evaluation process.

### 2.4. Description of the Sensor Prototype

The sensors were developed by the International Iberian Nanotechnology Laboratory (INL), located in Braga (Portugal), a member of the 4NoPressure project consortium, considering the textile interface under development by Impetus Portugal-Têxteis S.A. (IMPETUS). INL promoted the progress of material requirements and sensor models.

The hardware with individual sensors was developed, considering the design and modeling of devices, composed of micro- and nanomaterials, with properties of design and compliance. This prototype was subjected to a test phase for ergonomic evaluation related to comfort and convenience, considering user-centered design [26,27,28,29,30,31,32].

The selection of the materials used in the sensors was performed based on the requirements available in the literature, such as elongation and conformability [41,42], ultrathin [41], biocompatibility [41,42,43], biodegradability [41], self-degradation [41], conductivity [43], reliability [16], flexibility [42,43], mechanical strength [43], wash-resistant and durable [43,44], breathable [44], minimal comfort, grip, and life [42], lightweight [42,44], and bio-fluid [34]. These components are commercially available and have been used in other studies [43,45,46,47,48,49,50].

The target application of the prototype presented specificities regarding the shape and mechanical and physical characteristics of the sensors, namely flexibility with a high and low thickness (to minimize the topography when integrating the textile substrate). One explored approach used a polyimide (PII) substrate to build the pressure and temperature sensors. PII is a polymeric substrate compatible with microfabrication techniques for the production of thin-film sensors [51], such as tactile, pressure [52], and humidity sensors, being capacitive [53] or resistive [54]. Furthermore, the degree of biocompatibility and skin compliance [55] was also considered.

Regarding the transduction mechanisms explored and implemented in the prototype, resistance temperature detectors (RTDs) were selected for temperature assessment, while both capacitive and resistive sensors based on flexible membranes were selected for the pressure sensors. Resistive pressure sensors are based on contact area variation between two conductive plates (due to deformation) [56].

Different formulations of silicone elastomers were tested for direct attachment of the PII sensors to the fabric. Ecoflex was selected for its easy preparation and application and for the resulting mechanical characteristics, such as elasticity. PII is a humidity-absorbing material. Wrapping it in elastomers also confers protection by stabilizing the sensor properties [51].

As for the connection with the conductive fibers, besides the direct sewing on the contacts with the sensor and the use of conductive adhesives (epoxy with silver particles), a connection method using a flexible printed circuit board (Flex PCB) was also incorporated into the prototype. In this alternative, the electrical contacts of the thin-film sensor are not perforated by the stitching. Still, they are connected to standardized contacts on the Flex PCB using a tape with transverse electrical conductivity (also known as z-tape) while stitching is performed on the contacts of the Flex PCB [57]. These two approaches were implemented to comparatively evaluate their robustness and reliability.

The conductive wire used was a polyethylene terephthalate (PET) yarn, silver-coated using plasma technology, with 78 Dtex and a conductivity of 1.3 Ω/cm. To attach the Flex PCBs to the fabric, a thermoplastic mesh composed of aliphatic polyurethane ester was used [58]. Figure 2 illustrates the sequence of stitching the Flex PCBs, the assembly of the sensors on the Flex PCB, and the stages and areas of the Ecoflex application. The textile substrate and the conductive yarn, as well as the thermal fixation of the Flex PCBs to the fabric and the stitching, were selected by the Institute for Innovation in Fiber and Composite Materials (FIBRENAMICS).

The textile substrate used was a jersey-knitted fabric with 0.74 mm of thickness and 281 g/m^2^ of mass per unit area. Contact with the skin was made through a mixture of cotton yarns with Outlast viscose and, on the external side, a mixture of polyamide (PA) and modal (MO). The choice of using Outlast viscose is justified because the Outlast technology is distinguished by having incorporated microcapsules with phase-changing materials (PCMs) into its fibers, which contribute to the thermophysiological comfort [59].

### 2.5. Data Collection Instruments and Procedures

Data collection took place in April 2022. The prototype’s quality indicators in terms of the suitability of the Flex PCB and Ecoflex coated sensors were assessed using a questionnaire developed by the research team. Consideration was given to the physical and comfort properties and the suitability of the embroidery in relation to its limitation, flexibility, and comfort, which are potential risks when in contact with the skin. The theoretical rationale was based on published scientific studies on the biomechanics of medical devices and their cause/effect on the incidence of PIs, with questions regarding stiffness, roughness, imprinting, comfort, and size [60,61,62,63].

As this is a primary prototype to be evaluated and validated, new iterations will be subsequently evaluated by participants until a consensus is reached in order to develop a final version. The prototypes were divided into three main parts for evaluation: Prototype A: PII sensor coated in a protective layer of Ecoflex + embroidery; Prototype B: polyimide sensor coupled with Flex PCB + embroidery; Prototype A and B: specifically, regarding only the embroidery, which is not evaluated regarding its dimension because there was no final prototype to be tested that determines the evaluation of its size on the clothing (Figure 3).

Questionnaires were rated on a five-point Likert scale (“1—inadequate”; “2—slightly adequate”; “3—moderately adequate”; “4—adequate”; “5—very adequate”). The participants completed this first stage independently, with the prototype available for viewing and evaluation.

The FG technique was used for data collection, in which nine participants discussed and inquired about the usability and sensory perception of the prototype component presented as to its physical characteristics and the care settings considering the target population, as well as its advantages and disadvantages [64]. Nevertheless, considering health professionals’ wide range of performance areas, the contextual factors evoked from each report revealed significant variations in the opinions of the interviewees, which would not be possible to obtain with only the “QUAN” study [65,66].

In the “QUAL” methodology, due to the evaluative objective of the research, the interview guide included open-ended questions in random order, according to the discussion and the opinions being structured and adjusted to the prototype evaluation objective [65,66]. The questions scored were as follows: “Regarding the characteristics of each sensor, which aspects assessed in the quantitative stage do you think should be improved and why?” “What is the contribution of the biomechanics of medical devices to the increase in the incidence of PIs?”

The interviews were guided by the main moderator and assistant moderator, who wrote field notes and posed some complementary questions, using the focused interview approach. The moderators allowed exploring the findings that emerged with the group discussion and observations, generating new insights and questions about the evaluated product [65]. The FG speeches were recorded with an Olympus professional audio recorder, model WS-550M.

### 2.6. Data Analysis and Treatment Procedure

Data were analyzed using the Statistical Package for the Social Sciences (SPSS) software, version 24 (SPSS Inc., Chicago, IL, USA). To obtain results on the quality of the indicators, descriptive analysis was conducted to determine the mean scores of the items assessed. The analysis also integrated the evaluation of the confidence interval at 95% and the standard deviation (SD), which allowed calculating the coefficient of variation, obtained by dividing the SD by the mean of responses and multiplying it by 100, which is a standardized measure of dispersion that determines the precision of the distribution and the variability of responses, for which values below 20% are considered ideal [40]. Kendall’s coefficient of concordance was used for the ordinal Likert scale classifications. Results above 0.7 were considered a high correlation of concordance between the evaluators [67], and the value of *p* < 0.05 was established as statistical significance for all analyses.

The data from the “QUAL” evaluation were transcribed using MAXQDA software version 20, read in full, and analyzed with the Discourse of the Collective Subject (DCS) technique by mixing them with data from the first analyses extracted from the “QUAN” phase. On reading the transcribed speeches, excerpts relating to key expressions (KEs) were categorized into central ideas (CIs), and prior connections were made according to the elements assessed in the questionnaire. It should be emphasized that in the DCS technique, the CIs are interconnected to the KEs and not to the interpretations of the discourse, which makes up the first three steps of the DCS procedure [68,69].

Similar or complementary CIs were identified, and the KEs relating to the CIs were linked to the categories and coding in the text. The individual speeches were summarized and combined according to the researchers’ interpretations to form the speech of the collective subject [69,70]. For the integration of the “QUAN + QUAL” methods, the Joint Display approach [40] was used to present the results.

### 2.7. Ethical Aspects

This study received a favorable review by the Ethics Committee of the UICISA: E of the ESEnfC issued under number 701_07/2020. The participants signed the informed consent form (ICF), which explained the objectives and sequential steps of the study, in two copies of equal content. To preserve anonymity, the participants’ reports were identified by the letter E (nurse), followed by the order in which the reports and opinions occurred (e.g., E-01). After the transcriptions, the FG audio recordings were destroyed.

## 3. Results

Nine nurses participated in the study, with a mean age of 32.66 ± 6.28 years and a mean of 10.88 ± 6.19 years of nursing practice. The majority were women (55.6%), had a master’s degree (33.3%), and were specialists in rehabilitation nursing (33.3%) (Table 1). All participants were scientific researchers.

Prototype A, with PII covered by a protective layer of Ecoflex + embroidery, presented lower mean values for stiffness (1.56 ± 1.01) and roughness (2.11 ± 1.17), showing the lowest adequacy of the prototypes. Prototype B, with the PII sensor component coupled with Flex PCB + embroidery, presented a lower adequacy in stiffness (3.00 ± 1.22) and dimension (2.33 ± 0.50). Prototype A and B, which specifically evaluated the embroidery, revealed lower mean values in stiffness (1.88 ± 1.05), roughness (2.44 ± 1.01), and comfort (2.55 ± 1.42) (Table 2).

According to the DCS, friction and prolonged bed rest can generate PI risk associated with the sensor-based device. In the “QUAL” phase, the DCS complemented the specifications of the evaluations, making stiffness and roughness the items with the lowest adequacy (Table 3). Depending on the clinical context of the bedridden people and/or people with RM, these implications make the sensor prototype presented unsuitable or poorly suitable for the clinical context under discussion. The discourse on imprinting or marks that the device may cause on people’s skin revealed a KE related to the possibility that alternating decubitus position could cause imprinting due to friction caused by sheets sliding (Table 3).

The participants also discussed new configuration proposals, such as the use of liners over the sensors so that they do not come into direct contact with the skin. These aspects are associated with clothing design and user safety and resulted in low adequacy regarding the physical characteristics of the evaluated device. However, the comfort of the prototype sensors was also considered regarding the practicality of the garment and its usefulness in the provision of clinical care in hospital settings, long-term care, or home care (Table 3).

Embroidery also obtained low values regarding its adequacy as it became uncomfortable because of its lumpy texture. The dimension of the prototype sensors obtained low average values, and in the DSC, questions emerged about the width of the garment and its practicality in changing decubitus position, the anatomical region where the sensors would be placed, and the patient’s clinical needs related to mobility and the anthropometric interface (Table 3).

Table 4 shows the values of the agreement tests of the participants’ assessment, with a Kendall’s W of 0.359 (*p* = 0.039) for roughness and 0.318 (*p* = 0.022) for imprinting without sliding, interpreted as a moderate concordance among the participants. The other items presented a low coefficient of concordance.

## 4. Discussion

This study assessed the usability of one sensor-based prototype (Prototype A and B) designed for PI prevention in people with reduced mobility and/or bedridden patients. The results indicate that Prototype A, which has a PI prevention component covered by a protective layer made of Ecoflex, presented lower mean values of stiffness and roughness, indicating minor suitability. Prototype B, which has a PI prevention sensor component coupled with Flex PCB, showed a lower degree of suitability in terms of stiffness and dimension.

The DCS analysis indicated that friction and prolonged bedrest can generate the risk of PIs associated with the sensor-based device. The DCS complemented the specifications of the evaluations, and it was referred to in the discussions that the prototype was stiff and rough. These implications make the prototype unsuitable for clinical use in the context under evaluation.

The use of commercially available nanomaterials and components indicates that the developed prototype has potential for practical applications. However, further studies on the ergonomic properties of pressure, temperature, and humidity sensors in a single monitoring system are needed. After adjustments, the evaluated prototype will allow for the simultaneous monitoring of these parameters, providing a more complete picture of the observed conditions. Integrating sensors into a single monitoring platform will save space, reduce the number of necessary components and their complexity, and make the system more reliable and cost-effective.

Combining these materials and components resulted in a better understanding and design changes. The use of biocompatible, biodegradable, and self-repairing materials highlights the focus on the safety and sustainability of this evaluated prototype. Table 5 presents the minimum requirements in the literature and the materials used in the pressure, temperature, and humidity sensor prototype developed by the 4NoPressure project.

The prototype evaluated in this study consisted largely of polymer substrates. Prospects presented in the literature point to developing polymeric materials with sufficient mechanical properties to withstand friction and shear caused by body movements and/or alternating decubitus [61,71,72].

The versatility of synthetic polymers benefits from mechanical and hydrophobic adjustments for obtaining tough, malleable, waterproof, and/or breathable products at low cost. This diversity also enhances the creation of sensors with properties shaped to the broad physical and chemical functionalization, contemplating the skin microclimate [73,74,75].

Prototype A, with PII covered by a protective layer of Ecoflex, presented the lowest average values of stiffness and roughness, indicating the lowest suitability. The manufacturing method may be responsible for this result. There is limited evidence on using PII in sensors for PI prevention adhered to specific textiles [11,70].

The Ecoflex technology was chosen because it is widely used in other sensors and has excellent mechanical and piezoresistive properties [70]. These properties increase the durability of the device covered by Ecoflex, especially when considering exposure to body fluids and washing clothes [11].

This aspect underscores the importance of considering the end-use environment of a product when selecting materials and designing it. Sometimes, a material that performs well in a controlled laboratory environment may not be suitable for practical use in real-world conditions [11,70].

A study in China developed graphite-based strain sensors coated with Ecoflex. The authors used the substrate for better sensitivity and signal protection, which showed better detection capability and stability after several laboratory experiments. In the clinical context, these have only been used to monitor movement and breathing, and ergonomic comfort requirements have not been explored [76]. Regarding mechanical properties, flexibility and size are specific features that integrate the adequacy of ergonomic comfort. The device’s size may become inadequate due to postural changes and increased risk of PIs caused by skin contact pressure with the developed device [41,42,43,44,77].

McNeill’s study [77] used similar models to measure temperature, humidity, and pressure in a small and flexible size. The study highlighted the advantages of the device in terms of size and durability. The authors did not provide information on how the device should be applied to the skin, nor did they address any issues related to the manufacturing process and cost [77].

In an industrial context, studies recommend considering the manufacturing techniques of the substrates, which must follow a specific method for wearable sensors [71,72]. This is because the way substrates are manufactured can affect their performance and durability, as well as their adaptation to different environments and conditions of use [77,78].

In the context of user-centered design, the results can identify issues related to current manufacturing methods, which may not be optimized for the specific needs of the user. The evaluation uncovered that the prototype may be uncomfortable or poorly adjusted, requiring a revision of the project specifications and a reevaluation of the user’s requirements.

Based on the requirements in Table 5, elastomers were selected for use in the prototype development due to their mechanical, electrical, and structural properties. RTDs and elastomers can provide a robust solution for measuring temperature in challenging environments [56]. Covering an RTD with an elastomeric material can protect the sensor from environmental factors such as humidity, dust, and other contaminants. This can help to ensure that the RTD operates correctly and provides accurate temperature measurements over time [56,79,80,81].

The use of elastomers to cover RTDs in temperature sensors can improve the durability and accuracy of the sensor while also providing user-centric ergonomic comfort. By designing sensors with user comfort in mind, manufacturers can create products that are more appealing to end-users and more likely to be adopted in various applications [80,81,82,83].

Shintake [83] described the advantages of using elastomers, which promote greater elasticity, low friction coefficient, and better deformation. Recent studies [80,81,82] reported similar results, highlighting the detection capability and low manufacturing cost.

The classification of the dimension as “slightly adequate” suggests that the evaluators have some concerns about the size of the device and its potential to cause discomfort. The device’s design should consider the needs, preferences, and comfort of the user. By involving users in the design process and obtaining feedback on the device’s size, form factor, and overall usability, manufacturers can ensure that their devices are accurate, effective, and convenient to use [83,84].

The device size can also be optimized to balance the need for accuracy and sensitivity with user comfort. A study conducted in China reported the development of topological materials that can be embedded in cotton fibers [85]. A similar process is under development by partners of the 4NoPressure project, addressing the need for high sensitivity to signal performance.

The materials used, mostly composed of polymer substrates, are based on the possibility of modification, considering the versatility of synthetic polymers. These characteristics benefit mechanical and hydrophobic adjustments for obtaining resistant, flexible, impermeable, and/or breathable products at a low cost [74,75,85].

Prototypes A and B received criticism from nursing professionals regarding comfort and the need to reduce stiffness and roughness. These prototypes have a Flex PCB, which is highly conductive and composed of metallic and derivative materials, making them less flexible and more uncomfortable [84].

The study addressed the low performance of flexible circuits using Flex PCBs for operational reliability due to the movement of the human body. The study pointed out important technological advances for the development of components, considering flexibility and performance, manufacturing process, and automation [86].

Liman emphasizes the discomfort that the properties of some conductive materials cause on the skin, highlighting the need for intelligent textile manufacturing to meet the requirements for activating electronic components with the application of precise different materials [84]. The results of this study increase the importance of these data and the technological advancement to produce smart textiles for the prevention of PIs.

Researchers from Pakistan used a commercial sensor model. The PCB system was encapsulated with plastic and adhered to a wearable device to determine pressure points in the gluteal region through electrical stimulation. The authors concluded that the model is biocompatible and offers important advances for smart clothing for PI prevention [87]. Thus, the user-centered design emphasizes the need for prototype improvement, considering that the literature presents devices with similar objectives but different comfort characteristics.

Studies have shown that the decision to use specific materials to develop sensors directly impacts their medical properties, interfering with their biocompatibility [81,82,88,89]. Proposals to mitigate the use of uncomfortable materials derive from the careful choice of materials used in substrates to ensure the flexibility of wearable sensors [90] and elevate patient safety [91].

By analyzing the study results and applying user-centered design principles, it is possible to develop new strategies to improve technology in creating sensors for PI prevention. The insights obtained with this methodological approach provide valuable information for researchers, engineers, and healthcare professionals interested in developing PI prevention technologies [92].

A Chinese study reported the use of the traditional user-centered design approach for the development of a wearable device for elderly care. The authors concluded that the methodology is beneficial in terms of problem-solving for nursing professionals. This approach can ensure that devices meet their specific needs and enable them to perform their tasks efficiently and safely [93].

Prototype A and B with specifically evaluated embroidery showed lower average values of stiffness, roughness, and comfort. These results are important for the user-centered design issue. There is evidence of the possibility of developing clothing where modifications of the fabric substrate and woven electronics can be alternatives for recording signals from a type of conductive fabric [11].

The results of the study developed by Ye [94] showed relevant advantages of manufacturing textiles with conductive properties. Textiles can reduce the occurrence of PIs by having a smooth surface and high permeability. The study adds the high sensing capability of sensors integrated into the conductive fabric to pressure signals [94]. The use of conductive fabric with sensors can be an important way to reformulate the prototype evaluated in this research, allowing for the creation of more comfortable and discreet devices for PI prevention.

In addition, the use of new textile technologies can create more efficient and precise sensors, contributing to improving the quality of devices for preventing PIs. Conductive fabrics with sensors and other textile technologies can open new opportunities for research and development of PI prevention devices adaptable to user needs.

PIs are a common and serious problem that can occur in bedridden patients and lead to complications such as infections, pain, and longer hospital stays. The prototype of sensors to monitor pressure, humidity, and temperature, evaluated in the study, can be a promising solution to detect early signs of PIs and prevent their development [4,7,9]. By providing a mixed-methods study, the article offers a comprehensive overview of basic usability requirements related to ergonomic comfort [38,39].

The study is relevant to public health as it contributes to developing wearable and non-invasive technology that can detect pressure injuries in bedridden patients, providing opportunities for better health outcomes. It also provides information about the experiences and opinions of end-users who would use the technology and how they perceive the usability and effectiveness of the technology, as well as greater addition in daily use [95,96,97].

The results of a user-centered design study can be used to develop artificial intelligence (AI) algorithms to improve the design and functionality of products, including those that incorporate pressure, temperature, and humidity sensors. By understanding the needs and preferences of end-users, AI algorithms can be programmed to make decisions that prioritize the user experience and satisfaction [98,99,100].

This study is innovative because it utilizes new sensors that are specifically designed to meet the physical and comfort needs of users for preventing PIs. The study can contribute to promoting new research on the interactions between humans and the environment. The development and evaluation of new technologies, such as the clothing-based sensor, are gradually increasing in current research, and the mixed-methods study provides a comprehensive analysis of the potential impact of the technology on public health.

The development of clothing for PI prevention will be evaluated in terms of design and undergo pre-clinical and clinical tests with end-users. These tests will evaluate the overall characteristics of the clothing in terms of comfort, and specific tests will assess the functionality and conductivity of the sensors.

Therefore, the study addressed specific goals, such as material and user requirements, as well as usability, which was evaluated based on the needs of end-users to increase the adoption and acceptance of the platform. Collecting user feedback during the design and development process can help identify problems and areas for improvement. Moreover, ensuring that the sensor platform for PI prevention meets user requirements can contribute to cost-effectiveness.

This research is limited to the evaluative critique of one process as it is still in development, which reduces its generalizability and prevents the generation of case studies of the evaluated items. This limitation is based on clinical tests that have not yet been conducted and robust observations of the components of the target population. Thus, future studies to evaluate the performance of the sensor platform through testing in various environments and use cases can help ensure that it functions as expected and meets the intended specifications for accuracy and reliability.

## 5. Conclusions

It was observed in the evaluation of Prototype A that stiffness (1.56 ± 1.01) obtained lower averages, considered inadequate. The roughness (2.11 ± 1.17), imprinting without (2.22 ± 0.66) and with sliding (2.33 ± 1.00), and dimension (2.33 ± 0.50) were evaluated as slightly adequate. As for comfort, the evaluation results showed that it was moderately adequate (3.22 ± 1.48).

The evaluation of the components of Prototype B presented moderately adequate results regarding stiffness (3.00 ± 1.22), roughness (3.22 ± 1.09), imprinting without (3.22 ± 0.83) and with sliding (3.44 ± 1.01), and comfort (3.11 ± 0.92), values higher than Prototype A. Its dimension was evaluated as slightly adequate (2.77 ± 0.83). Prototype A + B + embroidery, which was used to evaluate the guide wire, was evaluated as inadequate regarding stiffness (1.88 ± 1.05) and slightly adequate regarding roughness (2.44 ± 1.01), imprinting without (2.66 ± 1.41) and with sliding (2.77 ± 1.30), and comfort (2.55 ± 1.42).

End-user acceptance scores were considered poor, given the materials used and the mechanical characteristics of the produced prototypes. These scores were especially significant considering the physical design of the prototype and the prospective user comfort. The stiffness and roughness of the plates and the conductive wire (embroidery) were considered inadequate, given the risk of PI development in bedridden people and/or people with RM, which is in line with the intrinsic clinical characteristics of the target population.

Our results confirm the need for substantial prototype development to ensure the development of comfortable, safe, and clinically reliable sensor-based clothing for people with RM and/or bedridden people.

In this way, given the problems associated with the incidence of the pathology of pressure injuries in the population, the care provided is increasingly personalized and these developments will become progressively faster. Thus, the establishment of multidisciplinary teams is essential, as is the involvement of end-users in developing and improving prototypes that can minimize the occurrence of pressure injuries.

## Figures and Tables

**Figure 1 ijerph-20-04039-f001:**
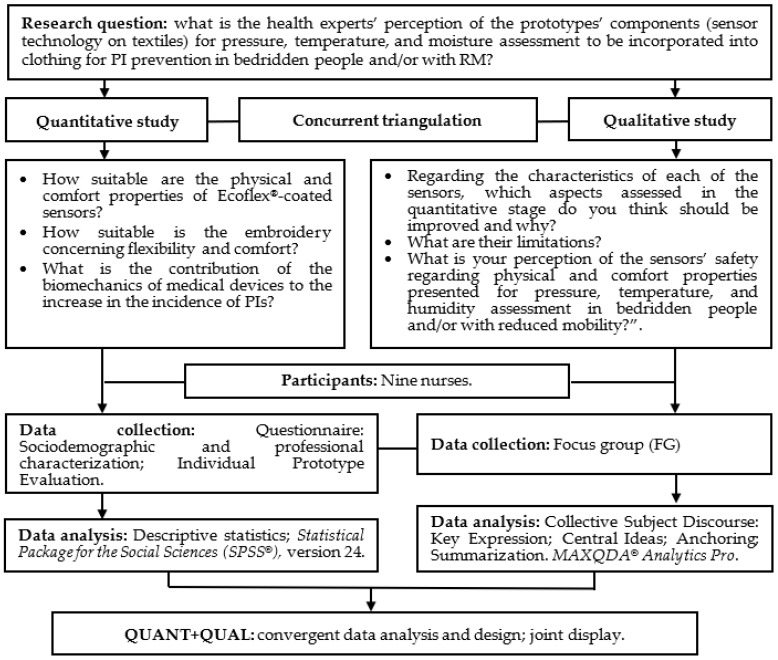
Representative diagram of the mixed method study design. Coimbra, Portugal, 2022. Note: QUANT: quantitative; QUAL: qualitative.

**Figure 2 ijerph-20-04039-f002:**
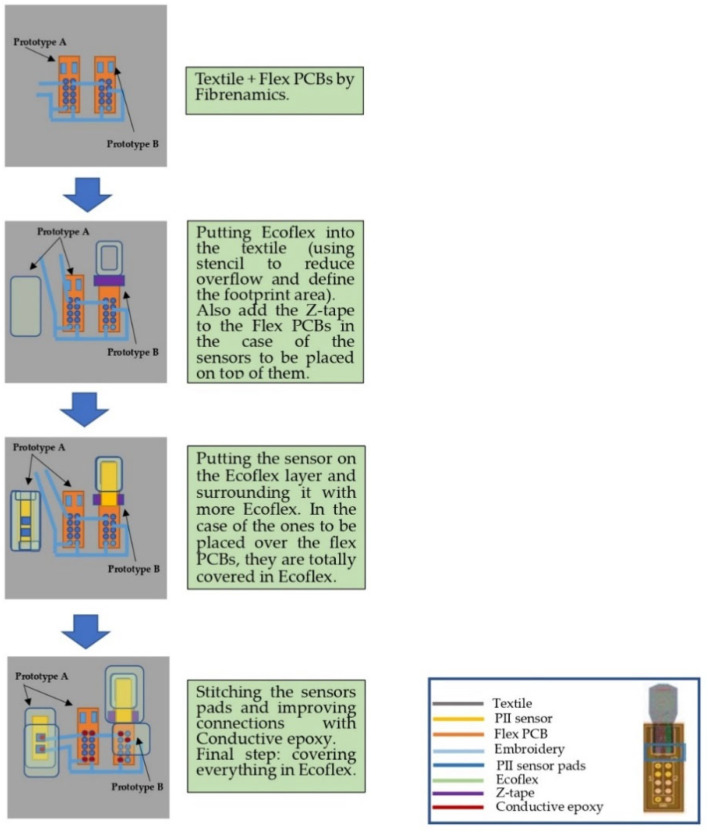
Integration of the temperature, pressure, and humidity sensors for expert evaluation of the physical and comfort aspects of the device. Coimbra, Portugal, 2022.

**Figure 3 ijerph-20-04039-f003:**
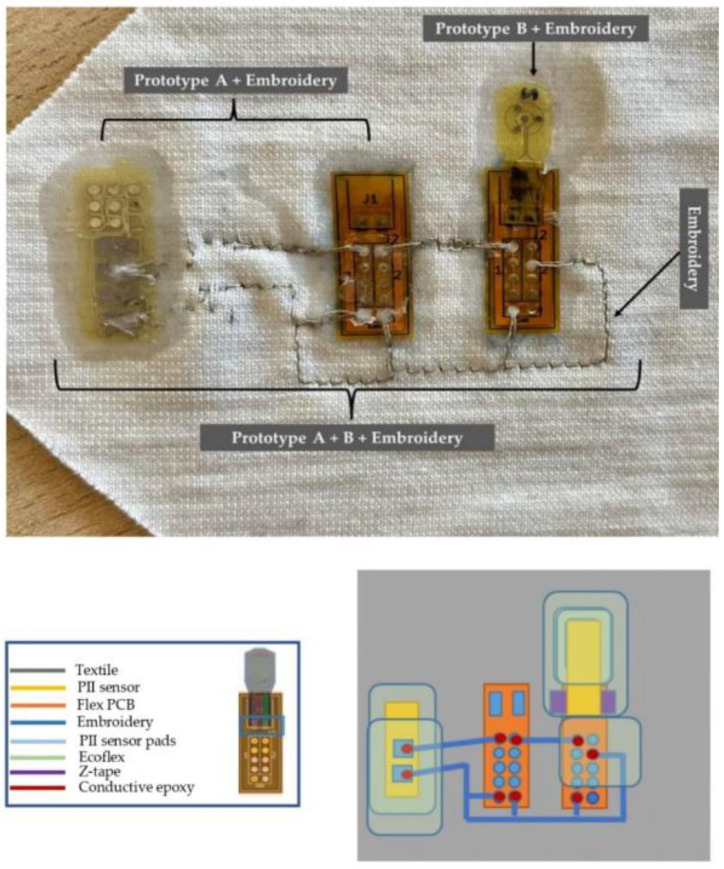
Prototype of the integration of temperature, pressure, and humidity sensors into the textile for evaluation by experts of the physical and comfort aspects of the presented device. Coimbra, Portugal, 2022.

**Table 1 ijerph-20-04039-t001:** Sociodemographic profile of the study participants who evaluated the sensor-based prototypes for PI prevention clothing. Coimbra, Portugal, 2022.

	M ± SD	
**Age**	32.66 ± 6.28	
**Years of experience**	10.88 ± 6.19	
	**N**	**%**
**Age group (Years)**		
20–29	2	22.2
30–39	6	66.7
40–49	1	11.1
**Gender**		
Male	4	44.4
Female	5	55.6
**Nursing Education**		
Undergraduate Degree	1	11.1
Postgraduation/Specialization	2	22.2
Master’s Degree	3	33.3
Doctoral Degree	2	22.2
Postdoctoral Degree	1	11.1
**Clinical practice area**		
Rehabilitation	3	33.3
Long-term care	1	11.1
Intensive care	1	11.1
General Nursing	2	22.2
Oncology	1	11.1
Medical–surgical	1	11.1

Note: M: mean; SD: standard deviation; N: total sample.

**Table 2 ijerph-20-04039-t002:** Distribution of the mean, standard deviation, and coefficient of variation of the study participants’ evaluation on all items of the sensors of Prototypes A and B and embroidery (n = 9). Coimbra, Portugal, 2022.

	**Prototype A + Embroidery**		
	**M ± SD**	**CV**	**CI**
**Stiffness**	1.56 ± 1.01	65.17	0.78–2.33
**Roughness**	2.11 ± 1.17	55.26	1.21–3.01
**Imprinting without sliding**	2.22 ± 0.66	30	2.08–4.36
**Imprinting with sliding**	2.33 ± 1.00	42.85	1.56–3.10
**Comfort**	3.22 ± 1.48	45.94	2.08–4.36
**Dimension**	2.33 ± 0.50	21.42	1.95–2.72
	**Prototype B + Embroidery**		
	**M + SD**	**CV**	**CI**
**Stiffness**	3.00 ± 1.22	40.82	2.05–3.94
**Roughness**	3.22 ± 1.09	33.92	2.38–4.06
**Imprinting without sliding**	3.22 ± 0.83	25.86	2.58–3.86
**Imprinting with sliding**	3.44 ± 1.01	29.43	2.66–4.22
**Comfort**	3.11 ± 0.92	29.83	2.39–3.82
**Dimension**	2.77 ± 0.83	30	2.13–3.41
	**Prototype A and B + Embroidery ***		
	**M + SD**	**CV**	**CI**
**Stiffness**	1.88 ± 1.05	55.8	1.07–2.69
**Roughness**	2.44 ± 1.01	41.47	1.66–3.22
**Imprinting without sliding**	2.66 ± 1.41	53.03	1.57–3.75
**Imprinting with sliding**	2.77 ± 1.30	46.86	1.77–3.77
**Comfort**	2.55 ± 1.42	55.72	1.46–3.65

Note: M: mean; SD: standard deviation; CV: coefficient of variation; CI: confidence interval; * embroidery dimension item was not evaluated, and the agreement coefficient represents the evaluation of Prototype A and Prototype B. Rating is performed on a 1–5-point scale, corresponding to inadequate and very adequate, respectively.

**Table 3 ijerph-20-04039-t003:** Joint display of quantitative and qualitative inferences summarized in the DCS, and meta-inferences based on the items of the sensor prototype evaluation. Coimbra, Portugal, 2022.

	QUAN Results	QUAL Results	Method Integration
	Structured Questionnaire	Summary of the Discourse of the Collective Subject	Metainferences
**Stiffness and roughness**	**Prototype A:** 1.56 ± 1.01; 2.11 ± 1.17 (low adequacy); **Prototype B:** 3.00 ± 1.22; 3.22 ± 1.09 (low adequacy); **Prototype A + B + embroidery:** 1.88 ± 1.05/2.44 ± 1.01 (low adequacy).	** *Friction (KE 48)* **	The stiffness and roughness of PCB Flex and other elements can damage the skin, which is usually fragile and prone to injury, especially by the time people with RM and/or bedridden people remain in bed and/or chair.
		*“It is too stiff, considering the clinical context in which the wearers may be admitted... If it is an intensive care unit, we will have problems. For example, in the case of older patients with unstable fractures who need to maintain bed rest. Friction and shear between the skin and sheets will influence the development of PIs [...].” (E-02)*	
		** *Contact time with the wearer’s skin (KE 18)* **	
		*“In hospitals, the time the device remains in contact with the patient may be much longer and the stiffness and roughness may constitute a problem, especially for people with reduced mobility. On days when you have to maintain rest, you may be unable to change position, rotate, massage [...].” (E-04)*	
**Imprinting with or without sliding**	**Prototype A:** 2.22 ± 0.66; 2.33 ± 1.00 (low adequacy); **Prototype B:** 3.22 ± 0.83; 3.44 ± 1.01 (low adequacy); **Prototype A + B + embroidery:** 2.66 ± 1.41; 2.77 ± 1.30 (low adequacy).	** *Alternating decubitus position can force imprinting (KE 21)* **	Considering the daily procedures of alternating decubitus in the hospital and/or at home, which can cause imprinting of the sensor on the skin and possibly PIs.
		*“If used with hospitalized people, people receiving long-term care, bedridden people, people who cannot move, alternating decubitus position can be a problem. People also use alternating pressure mattresses, which allow them to slide easily [...].” (E-07)*	
		** *A protective layer (KE 07)* **	
		*“The first sensor, the white one, has a protective layer on it that makes it softer, it is more comfortable. Their design is not the issue here, but their placement, to avoid printing, using some internally distributed pockets, lining, or similar. Using a lined pocket as a sensor distribution mechanism [...].” (E-02)*	
**Comfort**	**Prototype A:** 3.22 ± 1.48 (low adequacy); **Prototype B:** 3.11 ± 0.92 (low adequacy); **Prototype A + B + embroidery:** 2.55 ± 1.42 (low adequacy).	** *Practicality (KE 17)* **	The sensor materials showed resistance and roughness, and the wire conductor was considered an uncomfortable material and likely to cause skin injuries.
		*“Maybe pajamas help change the decubitus position and in various contexts. The use of alternating pressure cushions is becoming a daily procedure [...] If they use adequate mattresses, I would try to understand how these pajamas could help patients’ comfort [...].” (E-06)*	
		***Embroidery (KE 33)***	
		*“I could feel the embroidery knot. The conductive wire overlaps the fabric, even if on the reverse side of the fabric. The use is interesting, but the embroidery with the conductive wire on the other side felt uncomfortable.” (E-01)*	
**Dimension**	**Prototype A:** 2.33 ± 0.50 (low adequacy); **Prototype B:** 2.77 ± 0.83 (low adequacy).	** *Width (KE 38)* **	The size of sensors can increase the risk of injury due to their rigidity and roughness. The anatomical location of the sensors may imply greater matrix width for monitoring a certain area.
		*“It is very large, which is relevant in pajamas when you need to bend and change decubitus position [...].” (E-03)*	
		** *Location or anatomical points (KE 89)* **	
		*“We must consider this when choosing the anatomical region to place the sensor. If I place a twenty-centimeter sensor matrix in the sacrococcygeal region, will it give me the expected results without damaging the patient’s skin? [...].” (E-05)*	
		***Patient need (KE 57)***	
		*“It depends on the size of the whole pajama interface. The size of the sensor has implications on the other interfaces observed; as a participant, I honestly don’t know how much pressure per square meter there can be and if it is reasonably tolerated considering the patient’s clinical situation. The person can have very limited mobility or some changes in locomotion [...].” (E-07)*	

Note: *KE*: key element; PI: pressure injury; QUAN: quantitative; QUAL: qualitative.

**Table 4 ijerph-20-04039-t004:** Evaluation of the significance of the degree of concordance of the evaluation of the sensor prototypes (n = 9). Coimbra, Portugal, 2022.

	*Kendall’s W*	*X*^2^	DF	*p*	Concordance
**Stiffness**	0.359	6.467	2	0.039	Moderate
**Roughness**	0.248	4.467	2	0.107	-
**Imprinting without sliding**	0.318	11.458	2	0.022	Moderate
**Imprinting with sliding**	0.250	4.500	2	0.105	-
**Comfort**	0.111	2.000	2	0.368	-
**Dimension ***	0.111	1.000	1	0.317	-

Note: *Kendall’s W*: Kendall’s coefficient of concordance; *X*^2^: chi-square statistic; DF: degrees of freedom; *p*: statistical significance; * the embroidery item was not evaluated, and the coefficient of concordance represents the evaluation of Prototype A and Prototype B.

**Table 5 ijerph-20-04039-t005:** Prototype characteristics of pressure, temperature, and humidity sensors to prevent pressure injury. Coimbra, Portugal, 2022.

Properties	Physical Properties	Chemical Properties	4NoPressure	Advantages
Electrical	Electrical conductivity [43]	Reproducibility/Reliability [41]	Fully flexible printed circuit board (Flex PCB)	Studies define the advantages of fully flexible printed circuit boards
Mechanical	Flexibility [41,42,43,44]	Non-toxicity [41,42,43]	Ecoflex (easy preparation and favorable mechanical properties)	Biodegradable polymer compounds
	Mechanical resistance [41,42,43,44]	Resistance to washes [43,44]		
	Low coefficient of friction [43]			
	Thermoregulation [34]			
	Humidity management [34]			
Structural	Lightness (low mass per surface unit and low thickness) [42,44]			

## Data Availability

Data presented in this study are available upon request from the corresponding author. Data is not publicly available due to the device patenting process under development.

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
