# Peer review of "End-User Assessment of an Innovative Clothing-Based Sensor Developed for Pressure Injury Prevention: A Mixed-Method Study"

_ijerph, 2023, doi:10.3390/ijerph20054039_

Round 1

Reviewer 1 Report

There are excessive grammar and spelling mistakes: line 221-222 Analyses à Analysis, Line 155: 281 g/m2 à g/m2

It will be better if authors adjust one table on one page.

There is lack of experimental results of sensor. If results of electrical signals are included, that will be a better option.

The focus of paper is wearable textile sensors, however the uniform epoxy coating on sensor is not breathable. Is it not contradictory for the comfort applications?

Authors didn’t mention about the life time and washing cycles of sensors.

Author Response

Reviewer 1:

  • There are excessive grammar and spelling mistakes: line 221-222 Analyses à Analysis, Line 155: 281 g/m2 à g/m2.
    • Answer: The points described as spelling errors have been revised as requested.
  • It will be better if authors adjust one table on one page.
    • Answer: The tables were adjusted to fit on one page. As a result, some paragraphs had their position changed.
  • There is lack of experimental results of sensor. If results of electrical signals are included, that will be a better option.
    • Answer:  This suggestion was not accepted because it is not the objective of the study, which aimed to evaluate ergonomic and biomechanical comfort issues. The signal and conductivity tests will be evaluated on the final prototype of the lab-scale medical device.
  • The focus of paper is wearable textile sensors, however the uniform epoxy coating on sensor is not breathable. Is it not contradictory for the comfort applications?
    • Answer:  Some questions about the substrates used in the sensors were discussed between lines 189 to 195 on page 6 regarding biomechanics and comfort issues. It is pointed out that the use of the polyimide substrate was an industrial decision, and the evaluation encompasses this aspect as well. It should also be noted that, being an industrial decision, the use of epoxy was also referred to by the study participants and was classified as an outcome, subsequently discussed with the literature.
  • Authors didn’t mention about the lifetime and washing cycles of sensors.
    • Answer: The issue of sensor properties regarding lifetime and washout was not addressed. These evaluations are laboratory scale, and it was not the objective proposed by the study under evaluation.

I hope the adjustments are as requested.

We are at your disposal for any information about the study, based on the evaluation process.

My best regards.

Yours sincerely,

PhD. Anderson da Silva Rego

Nurse. Post-doctorate in Experimental and Applied Research in Care Technologies (TecCare)

The Health Sciences Research Unit: Nursing

Coimbra Nursing School, Portugal.

Reviewer 2 Report

This work mentions about End-users assessment of an innovative clothing-based sensor developed for pressure injury prevention: a mixed-method study. This study aimed to evaluate a prototype component of pressure, temperature, and humidity sensors to be integrated into clothing for the prevention of Pressure Injuries regarding physical and comfort requirements.

 The observations are as follows:

 1.      The novelty in this work is not clear. Try to highlight the same in the introduction section.

2.      This work needs to be compared with the literature.

3.      Is it possible to use AI technology for the prediction of risk based on the obtained data? The same can be applied on the large data set and opinions.

4.      The authors can work in this direction using some algorithms. Otherwise this work is simply acquisition of data using various sensors.

5.      The abstract and conclusion should provide the summary of work. The observations, outcomes, etc need to be mentioned. But, abstract and conclusion are not in this direction.

6.      Quantify abstract and conclusion for outcome observations.

Author Response

Reviewer 2:

  • The novelty in this work is not clear. Try to highlight the same in the introduction section.
    • Answer:  A paragraph was inserted in the introductory section for better understanding of the novelty and justification for conducting this study.

  • This work needs to be compared with the literature.
    • Answer:  A few paragraphs were inserted with a discussion of the literature in light of what was exposed in the results of this study.

  • Is it possible to use AI technology for the prediction of risk based on the obtained data? The same can be applied on the large data set and opinions.
    • Answer:  Paragraphs were inserted in the discussion about the possibility of using the results in artificial intelligence technology. The 4NoPressure project will develop a specific software to interpret the sensor signals, which may provide algorithms for this software to be able to learn by itself, from machine learning technology.
  • The authors can work in this direction using some algorithms. Otherwise, this work is simply acquisition of data using various sensors.
    • Answer:  Paragraphs were inserted in the discussion about the usability of the results and worked out using algorithms. It is worth pointing out that the results are not attributed to several sensors, but to a set of sensors of a prototype, built for a preventive garment for pressure ulcers. It is a technological innovation project.

  • The abstract and conclusion should provide the summary of work. The observations, outcomes, etc need to be mentioned. But abstract and conclusion are not in this direction.
    • Answer:  Summarized information was inserted, both in the abstract and the conclusion, about the results evidenced in this study.
  • Quantify abstract and conclusion for outcome observations.
    • Answer:  Summarized information was inserted, both in the abstract and the conclusion, about the results evidenced in this study.

I hope the adjustments are as requested.

We are at your disposal for any information about the study, based on the evaluation process.

My best regards.

Yours sincerely,

PhD. Anderson da Silva Rego

Nurse. Post-doctorate in Experimental and Applied Research in Care Technologies (TecCare)

The Health Sciences Research Unit: Nursing

Coimbra Nursing School, Portugal.

Reviewer 3 Report

The introduction part should be modified with recent research related to comfort evaluation. The authors want to evaluate the physical and comfort requirements of various sensors however related research is missing in it. 

The authors discussed the project "4NoPressure". However, reference related to it is missing.

2.4. is confusing and the type and preparation of sensors are not clearly understood.

Page 5, lines 175-180 are not clear. The third part (Prototype A and Prototype B) of the evaluation is not described clearly. why it is required to evaluate only embroidery if it is already added in previous parts?

The questionnaire is not described as related to discussed parameters.

Author Response

Reviewer 3:

  • The introduction part should be modified with recent research related to comfort evaluation. The authors want to evaluate the physical and comfort requirements of various sensors however related research is missing in it. 
    • Answer: A paragraph was inserted in the introductory section for a better understanding of the novelty and justification for carrying out this study and insertion of corresponding literature references.
  • The authors discussed the project "4NoPressure". However, reference related to it is missing.
    • Answer:  References from studies conducted under the 4NoPressure project have been inserted.
  • 2.4. is confusing and the type and preparation of sensors are not clearly understood. Page 5, lines 175-180 are not clear. The third part (Prototype A and Prototype B) of the evaluation is not described clearly. why is it required to evaluate only embroidery if it is already added in previous parts?
    • Answer:  The composition of the prototypes and the justification of only evaluating the embroidery was rewritten. This information can be seen starting on page 6. Names of the prototypes were inserted in Figure 2, 3 and Table 2.
  • The questionnaire is not described as related to discussed parameters.
    • Answer:  The description of the questionnaire in the method has been changed.

I hope the adjustments are as requested.

We are at your disposal for any information about the study, based on the evaluation process.

My best regards.

Yours sincerely,

PhD. Anderson da Silva Rego

Nurse. Post-doctorate in Experimental and Applied Research in Care Technologies (TecCare)

The Health Sciences Research Unit: Nursing

Coimbra Nursing School, Portugal.

Author Response

I hope the adjustments are as requested.

We are at your disposal for any information about the study, based on the evaluation process.

My best regards.

Yours sincerely,

PhD. Anderson da Silva Rego

Reviewer 4:

  • Thank you for providing this well-structured, well written and comprehensive paper. Focus of the paper is on the execution and evaluation of a structured questionnaire towards physical and comfort requirements of a smart textile system. Although, I am not an expert in this field, this part of the paper shows scientific depth and a well-structured approach. With respect to the hardware part and the textile implementation, only very little results are given. This is not a problem per se, because you choose another focus. However, when analyzing the quite bad questionnaire results and having a look at your probably quite large measurement set-up I am not surprised. As far as I understand, your approach is to measure humidity, temperature, and pressure at one spot. There are so many miniaturized solutions out there (even fiber-based or fiber-shaped) that you should defiantly have to motivate your chosen set-up more intensively and discuss it against other approaches.
    • Answer:  The project's goal is not to use miniatures already available on the market, but to develop specific sensors to measure temperature, humidity, and pressure. This is made clear in the methodology of the study.
    • The perception of end users is not always taken into consideration and the product available in the market may not have the range for which it was designed. In this sense, the results of this study aim to contribute so that this practice is integrated into the methodological process of technological products for use in the health area and does not set up delays in the construction process of sensorized clothing. The study fits into the perspective of TRL levels 1 and 2 of the Technology Readiness Level (TRL), and an objective description is provided in the introduction.

  • Figure 2: o Check the format of the labeling o Some additional information about positioning, size and connection of the sensors themselves would be nice - Page 5, line 175-180: o You talk about three parts for evaluation. Please name them as well as numbered variants (like variant/part I, II and III).
    • Answer: The names of the prototypes in the method, Figure 3 and Table 2 were changed. The additional information requested were the names of the prototypes. There is a legend for the characterization of the components of each prototype in the presented figure.

  • The point is that it took some time for me to realize that “Prototype A and B” is your 3rd part o from the description I could not get what is the qualitative difference between Prototype A and B from a mechanical/functional perspective: please go into more detail here o “Prototype A and B” if you had no final prototype, why implement it in the questionnaire anyway? And if you didn't have the prototype, what did you use? Please give more detail here as well and explain why your chosen way is appropriate.
    • Answer: Information has been inserted to differentiate the prototypes. It is mentioned that the prototypes are primary results, with indicators for possible changes. It was a validation study for further finalization according to the recommendations presented and development of the final prototype.

  • Figure 3: The actual quality of the given picture is not acceptable o Add information towards dimensions o Please add pictures of all 3 variants under investigation o Please add at least one picture of the device in the worn configuration (in contact with a test person) and discuss this set-up.
    • Answer: A new picture was inserted, with better quality, and the names of the prototypes were inserted.

Nurse. Post-doctorate in Experimental and Applied Research in Care Technologies (TecCare)

The Health Sciences Research Unit: Nursing

Coimbra Nursing School, Portugal.

Round 2

Reviewer 2 Report

The authors must highlight their work contribution clearly in the introduction.

The work needs clear statements like this hardware is new or already published and used in the paper so and so, algorithm new technique or already published in the work so and so. 

A comparison in the form of a table is needed to discuss what is so far available hardware in this regard. If already available in the literature then how this work is different from the existing work. Modifications, application aspects, cost, time and reliability all these things need to be compared with the literature work.

Author Response

Response to reviewers.

Corrections suggested by the reviewers for ijerph-2106592 - End-users assessment of an innovative clothing-based sensor developed for pressure injury prevention: a mixed-method study.

Greetings to the reviewer. We thank you for the important suggestions and we accept all of them as pertinent for the clarity of the study under evaluation.

The changes and arguments are described in topics and by reviewer.

Reviewer 1:

The authors must highlight their work contribution clearly in the introduction.

Answer: We thank you for your suggestions. We accepted them all and inserted paragraphs with the contributions of the study, as suggested.

The work needs clear statements like this hardware is new or already published and used in the paper so and so, algorithm new technique or already published in the work so and so.

Answer: We appreciate the suggestions. Paragraphs were inserted in the method on the composition of the prototype, which was developed with materials reported in the literature. In the discussion, paragraphs were inserted that discuss the use of materials and debate with the literature the main advances.

A comparison in the form of a table is needed to discuss what is so far available hardware in this regard. If already available in the literature, then how this work is different from the existing work. Modifications, application aspects, cost, time and reliability all these things need to be compared with the literature work.

Answer:  We accepted the reviewer's suggestions, which were very pertinent for a better understanding and clarity of the study. The table was inserted, as suggested. Subsequently, the main materials used in the development of the prototype and the advances on the theme were discussed with the literature.

I hope the adjustments are as requested.

We are at your disposal for any information about the study, based on the evaluation process.

My best regards.

Yours sincerely,

PhD. Anderson da Silva Rego

Nurse. Post-doctorate in Experimental and Applied Research in Care Technologies (TecCare)

The Health Sciences Research Unit: Nursing

Coimbra Nursing School, Portugal.

Reviewer 4 Report

Thank you for updating the manuscript. Now it has a more clear and understandable story.

Author Response

Greetings to the reviewer. We thank you for the important suggestions and we accept all of them as pertinent for the clarity of the study under evaluation.

I hope the adjustments are as requested.

We are at your disposal for any information about the study, based on the evaluation process.

My best regards.

Yours sincerely,

PhD. Anderson da Silva Rego

Nurse. Post-doctorate in Experimental and Applied Research in Care Technologies (TecCare)

The Health Sciences Research Unit: Nursing

Coimbra Nursing School, Portugal.